# Mechanisms Controlling MicroRNA Expression in Tumor

**DOI:** 10.3390/cells11182852

**Published:** 2022-09-13

**Authors:** Shipeng Chen, Ya Wang, Dongmei Li, Hui Wang, Xu Zhao, Jing Yang, Longqing Chen, Mengmeng Guo, Juanjuan Zhao, Chao Chen, Ya Zhou, Guiyou Liang, Lin Xu

**Affiliations:** 1Special Key Laboratory of Gene Detection and Therapy & Base for Talents in Biotherapy of Guizhou Province, Zunyi 563000, China; 2Department of Immunology, Zunyi Medical University, Zunyi 563000, China; 3The Second Affiliated Hospital of Zunyi Medical University, Zunyi 563000, China; 4Department of Medical Physics, Zunyi Medical University, Zunyi 563000, China; 5Department of Cardiovascular Surgery, Affiliated Hospital of Guizhou Medical University, Guiyang 550031, China; 6Department of Cardiovascular Surgery, Affiliated Hospital of Zunyi Medical University, Zunyi 563000, China

**Keywords:** miRNA, abnormal expression, tumor, regulatory mechanism

## Abstract

MicroRNAs (miRNAs) are widely present in many organisms and regulate the expression of genes in various biological processes such as cell differentiation, metabolism, and development. Numerous studies have shown that miRNAs are abnormally expressed in tumor tissues and are closely associated with tumorigenesis. MiRNA-based cancer gene therapy has consistently shown promising anti-tumor effects and is recognized as a new field in cancer treatment. So far, some clinical trials involving the treatment of malignancies have been carried out; however, studies of miRNA-based cancer gene therapy are still proceeding slowly. Therefore, furthering our understanding of the regulatory mechanisms of miRNA can bring substantial benefits to the development of miRNA-based gene therapy or other combination therapies and the clinical outcome of patients with cancer. Recent studies have revealed that the aberrant expression of miRNA in tumors is associated with promoter sequence mutation, epigenetic alteration, aberrant RNA modification, etc., showing the complexity of aberrant expression mechanisms of miRNA in tumors. In this paper, we systematically summarized the regulation mechanisms of miRNA expression in tumors, with the aim of providing assistance in the subsequent elucidation of the role of miRNA in tumorigenesis and the development of new strategies for tumor prevention and treatment.

## 1. Introduction

MicroRNA (miRNA) is a group of endogenous non-coding single-stranded small RNA that is widely found in eukaryotic organisms. It completely or partially combines with the 3′untranslated region (3′-UTR) of the target mRNA through base complementary pairing, which promotes the degradation of target mRNA or translational inhibition at the post-transcription level [1]. The first miRNA (lin-4) was discovered in *Caenorhabditis elegans* in 1993 [2], and so far, 38,589 human mature miRNA sequences have been annotated in the miRBase database. Under the appropriate time and environment, these diverse and abundant miRNAs are involved in regulating the normal expression of various functional genes to maintain the body’s homeostasis [3]. Recent studies have shown that the expression of miRNA is significantly different between tumor tissue and normal tissue-derived cells, and abnormal expression of miRNA is closely related to the occurrence and development of tumors [4]. Importantly, the restoration of miRNA expression is beneficial to the treatment of tumor patients, suggesting that it can be used as an important target molecule for tumor therapy [5]. Although most studies have focused on the regulation of downstream target genes by miRNA, the mechanisms of how tumor-suppressor/tumor-promoting miRNA is downregulated/upregulated in tumors are largely unexplained. Therefore, further elucidation of the regulatory mechanisms of miRNA expression in tumors is of great significance for the treatment and prognosis of tumor patients.

In this paper, we systematically expounded on the regulation mechanism of miRNA expression in tumors from the aspects of genetic mutation of miRNA genome, epigenetic change, aberrant RNA modification, abnormal splicing of processing enzymes, and regulation of long non-coding RNA (lncRNA), as well as extracellular secretion, so as to provide a basis for the development of clinical cancer therapy targeting miRNA.

## 2. The Production Process of miRNA

The classical pathway for the generation of mature miRNA mainly undergoes two sequential processes at different sites by the ribonuclease (RNase) type III enzymes DROSHA and DICER. In the nucleus, the miRNA gene is first transcribed into primary miRNA (pri-miRNA) by RNA polymerase II (Pol II) and undergoes capping, splicing, and polyadenylation [6]. Since then, the RNase III enzyme DROSHA and its cofactor double-stranded RNA binding protein DGCR8 participate in the formation of a so-called microprocessor complex to cleave pri-miRNA [7]. After processing in the nucleus, a stem-loop precursor miRNA (pre-miRNA) of about 70 nucleotides is generated and exported to the cytoplasm with the help of exportin 5 (XPO5) along with the guanosine-5′-triphosphate ras-related GTP-binding nucleoprotein [8]. After being released by XPO5, the RNase type III enzyme DICER performs a second shearing of pre-miRNA to generate mature miRNA duplexes [9]. Finally, such duplexes are loaded onto argonaute (AGO) proteins to form effectors called RNA-induced silencing complexes (RISC), and a strand is selected within them to become mature miRNA. Mature miRNA can bind to the 3′-UTR of specific mRNA to mediate target mRNA degradation, destabilization, or translational repression [10,11]. On the other hand, in addition to this classical way of production, miRNA can also be produced by some non-classical ways. For example, pri-miRNA can produce hairpin structures similar to pre-miRNA without the cleavage of DROSHA, indicating that the sources of miRNA are diverse [12]. 

Furthermore, newly published studies have shown that the transcription of miRNA is not completely carried out in the way of gene coding. The integrity of upstream regulatory sequences, the modified state of histones, the shearing of various processing enzymes, and long non-coding RNAs (lncRNAs) can all affect their normal expression and play the role in cancer inhibition or carcinogenesis, which shows the complexity of abnormal expression mechanisms of miRNA in tumors, as shown in Figure 1. Therefore, it is of great significance to further clarify the regulation mechanisms of miRNA expression in tumors.

## 3. MicroRNA Genomic Variation

### 3.1. Mutations in Promoter Regions of miRNA Encoding Genes

A promoter is a necessary sequence element for the initiation of gene transcription, which can combine with related transcription factors through specific sites and promote the transcription of target genes with the participation of Pol II [13]. In a variety of human malignant tumors, a major genetic variation type, SNP, has been observed in a large number of miRNA promoter regions and leads to abnormal expression of miRNA. For instance, SNP rs4938723 is located in the promoter region of pri-miR-34b/c, and this genetic variation increases the risk of rectal cancer [14] and renal cancer [15]. Further studies showed that in hepatocellular carcinoma (HCC), SNP rs4938723, which appears in the promoter of tumor suppressor gene miR-34b/c, may affect the binding ability with putative transcription factor GATA-X, thereby reducing the transcriptional activity of miR-34b/c [16,17]. In addition, the minor allele of SNP rs4705342 in the miR-143 promoter region can significantly enhance the binding ability of the transcription factor NF-κB and promote gene transcription, ultimately causing the upregulation of miR-143 expression. This leads to the downregulation of its target gene ORP8 and becomes an important factor for HCC in HBV-positive patients [18].

So far, many studies have demonstrated that miR-7 is a tumor suppressor, and the downregulation of miR-7 is usually closely related to the occurrence and development of tumors [19,20,21]. Based on the previous research progress, our team found that the -604 and -617 sites of the miR-7-2 promoter region in lung cancer tissues were mutated, and miR-7 expression with promoter site mutation was lower than that with non-mutated in tumor tissues. Besides, we further proved that mutation in the promoter region could significantly reduce the expression level of mature miR-7 [22]. These studies suggest that the genetic integrity of miRNA promoter sequences may induce the change of miRNA expression in tumors. 

At present, these studies mainly focus on the single SNP in miRNA promoter. However, whether multiple SNPs on the same sequence have different regulatory mechanisms or through coordinated regulation needs to be further elucidated in the future.

### 3.2. Variations of miRNA Coding Sequences

SNP in miRNA coding sequence (including pri-miRNA, pre-miRNA, and mature miRNA) can change miRNA processing, expression, or regulatory activity, thus leading to cancer. Le et al. [23] found that SNP in the pri-mir-146a sequence could affect the expression of miR-146a. This SNP could induce a new mGHG sequence at the apical junction of pri-mir-146a to interact with the double-stranded RNA-binding domain of DROSHA. Surprisingly, the orientation of the new mGHG sequence on pri-mir-146a was switched from basolateral to apical, causing the microprocessor to unproductively cut SNP-pri-mir-146a at its non-splicing site, eventually leading to the downregulation of miRNA.

In the Chinese population, SNP rs11614913 in pre-mir-196a2 and SNP rs3746444 in pre-mir-499 are associated with a significantly increased risk of breast cancer [24]. In addition, SNP rs3746444 in pre-mir-499 and SNP rs2910164 in pre-mir-146a are also significantly associated with the occurrence of cervical squamous cell carcinoma (CSCC) [25]. These carcinogenesis mechanisms may be achieved by altering the expression or processing maturation of related miRNAs. Importantly, for different environmental pressures, SNP at the same site may also regulate miRNA production through different processing mechanisms. Xu et al. [26] showed that the G allele of SNP rs2910164 in the stem structure corresponding to the pre-miR-146a sequence could cause the increased expression of mature miR-146a and was closely related to the risk of hepatocellular carcinoma. In contrast, Kogo et al. [27] reported that in gastric cancer, patients with GG genotype of the same SNP (rs2910164) had significantly lower expression levels of miR-146a than patients with CC genotype, and caused the upregulation of EGFR and IRAK1, which might be due to the diversity of molecular functions and processing patterns in different tumor types.

In addition, it has also been reported that, albeit rarely, mutations in mature miRNAs can lead to impaired tight regulation of target mRNA expression and contribute to poor patient outcomes. For example, SNP rs11614913, located in the 3p mature miRNA region of hsa-mir-196a2, is closely associated with the survival time of individuals with non-small cell lung cancer (NSCLC) [28]. Interestingly, this SNP also contributes to cancer susceptibility in a variety of cancers by impairing the ability of hsa-mir-196a2-3p to regulate downstream target genes, or by changing the type of regulated target genes [29]. *Analogously*, this phenomenon also was confirmed by Kawahara et al. [30], who found that variation in the mature miR-376 sequence caused by RNA editing was sufficient to redirect miR-376 to silence a new set of target genes.

In summary, the above studies have shown that the variation of the miRNA coding sequence in tumors can significantly affect the ability of miRNA processing, expression, and regulation through different mechanisms, which is depicted in Figure 2. However, the exact mechanisms by which these mutations occur in tumor cells need to be further elucidated by turning to genetics, environmental pressures, and tumor types.

## 4. Regulation of miRNA by Epigenetic Modifications

### 4.1. The Influence of DNA Methylation on miRNA Expression

DNA methylation is a DNA chemical modification that catalyzes the binding of methyl groups to the 5th carbon of cytosine by DNA methyltransferases, which is essential for regulating gene expression [31]. Two DNA methylation patterns were observed in cancer cells, including the increase in CpG island methylation in gene promoters and the decrease of global DNA methylation patterns [32]. Recent studies have shown that the methylation level in the CpG island of the miRNA promoter in tumor cells is significantly increased, especially in tumor suppressor genes, as shown in Figure 3A. Multiple miRNA loci, including miR-9-1, miR-193a, miR-137, miR-342, miR-203, and miR-34, were found to be abnormally elevated in methylation levels in human cancers [33,34]. Among them, miR-34a is involved in regulating the expression of genes related to cell cycle, differentiation, and apoptosis, and exerts a tumor suppressor function by reducing cancer stemness and increasing drug sensitivity [35]. The CpG island of the miR-34a promoter is hypermethylated, resulting in its expression being silenced in a variety of tumors, including breast, colon, lung, and bladder cancers [36]. Importantly, when the expression of miR-34a is restored in tumor models, many classic proto-oncogenes such as MYC, KIT, BCL2, and SIRT1 are strongly downregulated [37]. Similarly, Hashimoto et al. [38] observed the hypermethylation in the upstream region of miR-181c in gastric cancer tissues and cells. Treatment with 5-Aza-CdR, a methyltransferase inhibitor, could restore the expression of miR-181c and inhibit the growth and proliferation of gastric cancer cells, suggesting that miR-181c has a tumor suppressive function in gastric cancer cells and its expression is downregulated by DNA methylation.

Global DNA hypomethylation in tumors is considered the companion of genomic CpG island hypermethylation, but it usually appears in different sequences [39]. In fact, tumor cells generally exhibit more global hypomethylation of DNA than CpG island hypermethylation, resulting in a net reduction in genomic 5-methylcytosine content, which provides the possibility for tumor development [40,41]. Because global DNA hypomethylation can lead to genomic instability and induce aneuploidy, chromosome translocation, and copy number change [42,43]. It promotes tumor progression through abnormal gene expression, including miRNA, oncogenes, and so on [44,45,46]. For example, the hypomethylation of miR-106b, miR-25, miR-93, miR-23a, and miR-27a promoter in HCC leads to upregulation of their expression and facilitates the growth of hepatocellular carcinoma cells through various pathways [47].

For the mechanism of miRNA differential methylation in tumors and tissues, studies have shown that it may be related to DNA motif background and different cellular internal factors. DNA methylation depends on the local context of gene sequence, which is reflected in differences in the distribution of methylated motifs: hypermethylated motifs are mainly located in promoter regions rich in CG or CpG islands, whereas hypomethylated motifs are mainly found in gene bodies [48]. In addition, the expression of each isoform of DNMTs in tumors is regulated by multiple factors and they have different affinities for intrinsic DNA motifs, resulting in complex and diverse modifications and regulation of DNA methylation in the genome [49]. Thus, CpG islands that are unmethylated in normal tissues are methylated in tumor cells, and this flexible regulatory mechanism is exploited by tumor cells to meet their own needs. Furthermore, another regulatory mechanism that causes differential methylation modification may arise from the preference of specific transcription factors for methylation motifs in tumor cells [50]. A typical example is Krüppel-like factor 4 (KLF4), which binds preferentially to methylated sequences and activates rather than represses transcription of related genes and tumor development [51,52]. This specific selectivity can have a major impact on gene transcription, especially cancer-promoting miRNAs that are hypermethylated in tumors.

Interestingly, recent studies have shown that there is a feedback loop between miRNA and DNA methylation. MicroRNA can reverse regulate DNA methylation by targeting DNA methyltransferases or methylation-related proteins [53]. In short, abnormal expression of miRNA caused by the altered methylation level is widespread in various tumors, and the crosstalk between miRNA and DNA methylation, as well as the specific methylation regulators in tumors, may helpfully provide some new therapeutic targets.

### 4.2. The Influence of Histone Modification on miRNA Expression

Histone modification is a plasticity regulatory mechanism. During normal biological development, histone modifications need to be precisely regulated at each developmental stage. Once some stimulus-induced dysregulation of the regulatory program favors cell survival, this adaptive program will be infinitely amplified with the growth of the dominant cells [54]. The dysregulated landscape of histone epigenetic modifications in tumor cells is shown in Figure 3B.

#### 4.2.1. Histone Acetylation

As one of the most common epigenetic regulators, histone acetylation is involved in the expression of miRNA in tumors by regulating the transcriptional activity of genes. Studies have shown that multiple lysine residues in the histone tail can be modified with acetyl groups, which weakens the binding degree between negatively charged DNA and histone by neutralizing the alkaline charges at lysine residues [55]. An abnormal acetylation pattern of histone has been reported as a common feature of human tumor cells [56]. Due to the overall activation of DNA in tumor cells, this provides a large number of modification sites for acetylases, which promote gene transcription through histone acetylation. Such as, it has been shown that in the breast cancer cell line SKBr3, treatment by histone deacetylase (HDAC) inhibitors caused a rapid change in the expression levels of 27 miRNAs, indicating that histones are extensively regulated by acetylation modifications in tumor cells [57].

Notably, some transcription factors can also regulate miRNA expression through histone modifications. For example, Myc can directly or indirectly regulate the target genes of cell growth and proliferation to promote tumorigenesis [58]. Zhang et al. [59] reported that Myc induces histone deacetylation and histone trimethylation to inhibit miR-29 expression in B-cell lymphoma (BCL), as evidenced by Myc can recruit histone deacetylase 3 (HDAC3) and zeste homologue 2 (EZH2) to the miR-29 promoter to form C-Myc/HDAC3/EZH2 co-repressor complex, which leads to miR-29 transcriptional silencing [59]. In ovarian cancer, the cancer suppressor miR-99a reduces cell proliferation through the AKT/mTOR pathway and downregulates the expression of its target gene HOXA1 to inhibit epithelial–mesenchymal transition (EMT). However, the transcription factor YY1 can attenuate the anti-tumor function of miR-99a and promote the dryness of ovarian cancer (OC) cells [60]. Mechanistically, YY1 reduces the acetylation level of miR-99a by recruiting HDAC5 (histone deacetylase 5) to the miR-99a promoter, and finally inhibits the expression of miR-99a [60,61].

These studies have shown that the altered expression of miRNA is partially regulated by histone acetylation in tumors. However, the exact cause remains elusive.

#### 4.2.2. Histone Methylation

Accumulated evidence suggests that the disruption of the balance of histone methylation and demethylation can lead to abnormal expression of related genes, including miRNA, and ultimately contribute to tumor progression [62,63]. Lysine at different sites of the N-terminal tail of histone can be methylated and produce different biological effects, which indicates the complexity of the regulation mechanism of histone methylation. For example, H3K4me3 and H3K36me3 are involved in the transcription and expression of active genes. While H3K9me3, H4K20me3, and H3K27me3 contribute to gene silencing [64]. In diffuse large B-cell lymphoma, upregulated miR-193b and miR-365 are associated with the deletion of H3K27me3 and enrichment of H3K4me3. Conversely, the downregulated miRNAs, including miR-223, miR-150, and miR-451 are correlated with the enrichment of H3K4me3, indicating that histone methylation is the epigenetic mechanism of these differentially expressed miRNAs [65]. In addition, miR-139 is downregulated as a tumor inhibitor in many tumor types. In NSCLC, miR-139 and its host gene PDE2A are apparently silenced by H3K27me3, which is independent of promoter DNA methylation [66]. Knockout of EZH2 or treatment with HDAC can restore miR-139 expression and promote metastasis of lung cancer [66]. Consistently, miR-133a is silenced by epigenetic factors in lung cancer cells, leading to upregulation of its target gene PTBP1 and accelerated lung cancer cell growth and metastasis. Mechanistically, KDM5C, as a specific demethylase of histone H3K4, causes histone demethylation of miR-133a promoter to inhibit transcriptional activation of miR-133a [67].

Up to now, a large number of different types of histone methylating and demethylating proteases have been discovered. Furthermore, the methylation status of histones is highly dependent on the regulation of these different proteins. In mammalian chromatin, Suv39h histone methyltransferases (HMTases) selectively monomethylate H3K27 and trimethylate H3K9 [68], while KMT2A (also known as MLL1) is responsible for catalyzing the dimethylation and trimethylation of H3K4 [69,70,71]. Therefore, dysregulation of specific modifiers may directionally alter the transcriptional activity of some miRNAs. For example, EZh2 dysregulation in lymphoma causes an increase in H3K27me3 labeling [72,73]. This will lead to epigenetic silencing of a large number of miRNAs. Furthermore, in tumor cells, this dysregulatory mechanism may create a positive feedback to further promote tumor progression.

#### 4.2.3. Other Modifications of Histones

In addition to these well-known modifications, histones can be modified in other ways and jointly determine the structure of chromatin. These modifications include phosphorylation, ubiquitin, citrullination, butyrylation, hydroxylation, formylation, propionylation, and crotonylation [74,75,76]. To some extent, the regulatory role of these histone-modified markers in various cancers has not been fully elucidated. Recently, however, there is still some evidence that they participate in tumor progression by regulating miRNA expression. For example, the phosphorylated histone H2AX binds to the miR-3196 promoter, which leads to the transcriptional silencing of miR-3196 by increasing H3K27 trimethyl in the promoter region of miR-3196 and inhibiting the binding of RNA Pol II [77]. In lung cancer cells, decreased phosphorylation of H2AX leads to high expression of miR-3196, which inhibits lung cancer cell apoptosis by targeting the p53-upregulated apoptosis regulator (PUMA) [77]. Furthermore, E3 ligase HectH9, which mediates ubiquitin modification, induces K63 polyubiquitin modification of DDX17 under anoxic conditions and promotes its dissociation from the pri-miRNA-Drosha-DCGR8 complex. This will hinder the processing and maturation of a large number of tumor suppressor miRNAs, including miR-16 and miR-34a, thus favoring the expression of cancer stemness genes [78]. All of this evidence suggests that miRNA expression is regulated by multiple histone modifications, and further exploration of the combinatorial effects of different histone modifications might be a key work in this field.

## 5. Regulation of miRNA by RNA Modification

Recent studies have revealed that alterations in the ncRNA transcriptome may be a novel mechanism of tumorigenesis, especially targeting the modification regulation of miRNA. Although the regulation of miRNA by chemical modifications such as m5C, hm5C, N6-methyladenosine (m6A), m1A, and uridylation has not been fully elucidated, RNA modification still provides researchers with a broad research prospect [79]. Most current studies have shown that m6A and uridylation modifications have a significant impact on the expression of miRNA and determine its fate in the occurrence pathway.

### 5.1. The Influence of m6A Modification on miRNA Expression

At the transcriptome level, as the most abundant RNA modification in eukaryotes, the role of m6A modification in regulating miRNA expression has become increasingly prominent [80]. The process of m6A modification is mainly coordinated by methyltransferase (m6A writer), demethylase (m6A eraser), and m6A binding protein (m6A reader) [81]. Direct evidence showed that in breast cancer cells, m6A methylation motifs are abundantly enriched in pri-miRNA sequences but absent in pre-miRNA sequences [82]. The m6A methylation motif provides a site for methyltransferase (METTL) to mark pri-miRNA with m6A, which facilitates the recognition and processing of DGCR8, suggesting that the regulation of miRNA expression by m6A modification occurs primarily during pri-miRNA processing and exhibits m6A dependence [82], as shown in Figure 4.

Most of the proteins involved in m6A modification belong to oncogenes, and they are frequently upregulated in tumor tissues, thereby increasing the processing and expression of oncogenes and promoting the malignant development of cells [83]. For example, the expression of METTL3 is significantly upregulated in bladder cancer, which enhances DGCR8 processing and promotes its expression by increasing the cancer-promoting miR-221/222 m6A modification [84]. Consistently, cigarette smoke induces METTL3 overexpression, which promotes the processing and maturation of miR-25-3p by increasing the level of m6A modification of pri-mir-25. Subsequently, high levels of mature miR-25/miR-25-3p inhibit the expression of PH structural domain leucine-rich repeat protein phosphatase 2 (PHLPP2) and induce activation of oncogenic AKT-p70S6K signaling to promote pancreatic cancer progression [85].

Although tumor suppressor genes processing may also be increased, differences in gene transcription levels already dictate that more oncogenes undergo m6A-modified processing than suppressor genes. In addition, another accepted mechanism may be that specific regulators of m6A modification act by blocking m6A modification of suppressor genes, which have been reported. After knockdown of the m6A demethylase FTO in HEK293 cells, several mature miRNAs were nevertheless downregulated [86]. In addition, the m6A reader HNRNPA2B1 has been shown to recruit DGCR8 after recognition of m6A to facilitate the shearing process of pri-miRNA by DROSHA. However, HNRNPA2B1 overexpression shows opposite effects on miR-222, miR-29a-3p and miR-29b-3p expression and leads to endocrine resistance in breast cancer cells [87]. All of this evidence fully demonstrates the complexity and diversity of m6A modification and the regulation of subsequent miRNA processing.

Current studies mainly focus on the effects of aberrant m6A modification, but the mechanisms of how various functional components involved in miRNA m6A modification are altered are not fully understood. Furthermore, how m6A modification differentially regulates miRNA processing and expression in tumor cells is unclear. Therefore, further studies on these molecules are necessary in the future to refine the mechanisms of miRNA expression regulation.

### 5.2. The Influence of Uridylation on miRNA Expression

The role of uridylation on miRNA was first identified in the precursor sequence of let-7 [88]. Uridylation of the 3′ end of pre-let-7 mediated by Lin28 evades DICER processing and induces pre-let-7 degradation [88]. Conversely, in human cells, modification of pre-let-7 monouridine by the terminal uridylyltransferase TUT7/4/2 can switch the optimal structure from a 1 nt 3′ protruding end to a 2 nt 3′ protruding end, thereby enhancing processing shear of DICER [89]. Interestingly, TUT, a broad RNA modifying enzyme, can also switch specific pre-miRNA from degradation mode to processing mode to promote miRNA expression, a switch that is determined by the cellular environment [90]. For example, in TUT4/7-depleted prostate and OC cells, some specific miRNAs showed differential expression patterns [91]. In particular, miR-200c-3p and miR-141-3p, which are diagnostic markers of OC, are significantly downregulated and can impair the migration ability in OC cells, suggesting that uridylation modifications regulating miRNA expression may have specific regulatory mechanisms in different tumor cells [91].

## 6. Abnormal Cleavage by Processing Enzymes

Although numerous studies have demonstrated that controlling the expression of individual miRNA has oncogenic or pro-cancer effects, the overall altered expression of miRNA in tumor cells compared to normal cells strongly suggests that the processing components of miRNA may be dysregulated in tumors [92]. Table 1 shows the outcome of dysregulation of some miRNA processing components in tumors.

### 6.1. Microprocessor Abnormalities and miRNA

In the nucleus, a heterotrimeric complex (called a microprocessor) consisting of one DROSHA and two DGCR8 molecules mediates the splicing of pri-miRNA. Among them, DGCR8 interacts with the stem and apical elements of pri-miRNA through its double-stranded RNA (dsRNA) binding domain and RNA-binding heme domain, respectively. Meanwhile, DROSHA acts as a “ruler” for measuring the 11-base distance of the single-stranded to double-stranded RNA (ssRNA-dsRNA) junction and cleaves the stem-loop of pri-miRNA to release pre-miRNA [112,113]. Besides, the microprocessor contains several functional cofactors, including DEAD-box, RNA helicase, p72 (DDX17), and the Ewing’s sarcoma family of proteins, which together promote the fidelity and activity of DROSHA processing [114]. However, abnormal expression or function of microprocessors and related components play important roles in tumors by altering miRNA processing and maturation to set the stage for cellular transformation and tumorigenesis in vivo [115,116,117]. One of its upstream mechanisms may be due to genetic mutation or transcriptional silencing/activation of a large number of processing components in tumors that disrupt the normal processing of miRNA. 

### 6.2. XPO5 Abnormalities and miRNA 

XPO5, as a “vehicle” of miRNA, shows paralysis or overload of transportation in cancer. There has been evidence that genetic mutations in XPO5 lead to defects in the C-terminal region of the pre-miRNA/XPO5/Ran-GTP ternary complex and retain the pre-miRNA in the nucleus [100]. In hepatocellular carcinoma, XPO5 is phosphorylated by ERK, leading to its conformational change by the prolyl isomerase Pin1, which ultimately reduces the ability of XPO5 to export pre-miRNA [118]. These studies suggest that the output mechanism of nucleoplasm may be damaged in tumor cells.

### 6.3. DICER Abnormalities and miRNA

When the pre-miRNA is transported to the cytoplasm, DICER participates in the final cutting process. DICER recognizes dsRNA and acts as a second molecular “ruler” by cleaving it at a specific distance from the end of the helix [119]. Consistently, mutations in the DICER gene are also frequently present in tumors and lead to impaired miRNA biogenesis and processing [120]. In addition, many regulatory factors regulate cancer progression by targeting the expression of DICER. For example, activated HIF-1α in tumors downregulates Dicer expression by inducing ubiquitination of the E3 ligase Parkin, and further reduces the expression of tumor suppressor miRNAs, including let-7 and miR-200b, which promotes EMT and metastasis in tumor-bearing mice [121].

In short, various processing enzymes and their components not only determine the correct production steps of miRNA but also maintain the normal expression of miRNA. Notably, the regulatory mechanisms of these processing steps are often different in various tumor types, and the expression patterns or functions of the same molecules may differ, so future perspectives are necessary to focus on tumor-specific targets for further exploration.

## 7. Regulation of miRNA by lncRNA

LncRNAs, including long intergenic noncoding RNAs (lincRNAs), circular RNAs (circRNAs), and pseudogenes, can regulate tumor biological behaviors through a competitive endogenous RNA (ceRNA) mechanism [122,123]. They act as a “sponge” for miRNA by sharing miRNA response element (MRE), thereby reducing the number of miRNAs that target mRNA, as shown in Figure 5.

### 7.1. Regulation of miRNA by lincRNA

LincRNA is an RNAi molecule consisting of hundreds to thousands of nucleotides that has been shown to play an important role in gene regulation. Although lincRNA has diverse regulatory mechanisms in tumors, their role as decoys for miRNA in the ceRNA regulatory network has become increasingly prominent. linc-ROR is upregulated in tumor tissues and promotes tumor progression by inducing EMT and promoting malignant abilities such as proliferation, migration, etc. [124,125]. Mechanistically, linc-ROR acts as a ceRNA of miR-205, inhibiting its expression and indirectly promoting the expression of miR-205 target genes [124]. Furthermore, linc-ROR also acts as a ceRNA of miR-145 and promotes the invasion of triple-negative breast cancer through the linc-ROR/miR-145/ARF6 pathway [126]. In contrast, lincRNA-p21 has oncogenic effects and can play a role in tumors by targeting different miRNAs, including miR-9 [127], miR-17-5p [128] and miR-181b [129]. Notably, some lincRNAs can also serve as feedstock for miRNAs that are processed to form mature miRNAs to participate in the biological behavior of tumors. For example, lincRNA H19 serves as a major source of precursors and a regulatable reservoir for miR-675 by regulating its processing and expression in response to cellular stress or oncogenic signals [130,131]. These studies suggest that the expression and function of multiple miRNAs in tumors are regulated by lincRNA and lead to altered expression of related target genes.

### 7.2. Regulation of miRNA by circRNA

CircRNAs are a class of single-stranded closed RNA molecules, usually formed by the reverse connection of exonic precursor mRNA, which indirectly regulate the activity of miRNA target genes by acting as miRNA sponges through abundant miRNA binding sites [132]. Recent studies have shown that the expression of circRNA is mainly expressed in specific cell types and tissues, and is dynamically regulated by various environments, suggesting that circRNA may play an important role in tissue development [133]. Moreover, its dysregulation can strongly affect the expression of miRNA and participate in tumorigenesis. The most representative of them is CIRS-7 (also known as CDR1as), which contains more than 70 conserved miR-7 target sites and is a super-sponge of miR-7 [134,135]. MiR-7 has been proven to inhibit cancer cell growth and promote apoptosis by directly targeting and downregulating key oncogenic factors in cancer-related signaling pathways [136,137]. In addition, our group further revealed that miR-7 plays an important role in alleviating the ConA-induced acute autoimmune liver injury model in mice by regulating immune cells. MiR-7 negatively regulates the activation and function of CD4^+^ T cells through the MAPK4 pathway, thereby reducing the pathological changes of autoimmune hepatitis [138]. This study enriches the regulatory function of miR-7 and provides researchers with a new perspective on the involvement of miR-7 in immune-related diseases including tumors. Notably, miR-7 was frequently downregulated in various tumor tissues, and the expression levels of miR-7 and CIRS-7 were negatively correlated. In tumor cells, upregulated CIRS-7 exerts its sponge effect by targeting miR-7 and upregulates its key target genes, including EGFR, CCNE1, and PIK3CD, to induce tumor cell proliferation [139,140]. 

The latest literature has documented that circ-Foxo3 has binding sites for multiple miRNAs and can act as a ceRNA to inhibit or promote tumor growth [141]. Specifically, circ-Foxo3 upregulates the expression of NFAT5 nuclear factor through a mechanism of sponge miR-138-5p/miR-432-5p to promote the proliferation and invasion of glioblastoma [142]. In gastric cancer, circ-Foxo3 upregulates the expression of USP44 by targeting miR-143-3p, thereby exerting a tumor-promoting effect [143].

In all, the above current studies have shown that circRNA, as a powerful miRNA inhibitor in tumors, is a novel marker for cancer diagnosis and potential therapeutic targets.

### 7.3. Regulation of miRNA by Pseudogene

Pseudogenes are a special group of lncRNA that develop from protein-coding genes and have been regarded as “evolutionary garbage” due to the loss of their ability to encode proteins [144]. However, recent studies have found that pseudogenes and their corresponding coding genes can bind to the same miRNA and thus act as a ceRNA, suggesting the importance of pseudogenes in the regulation of miRNA expression [145]. PTEN P1 is the first pseudogene found to regulate its parental gene PTEN through a ceRNA mechanism [146]. In tumors, PTEN P1 acts as a bait to adsorb and degrade miRNA targeting PTEN, and actively regulates the expression of miRNA targeting PTEN, such as miR-17, miR-21, miR-214, miR-19, and miR-26 families, resulting in upregulation of PTEN gene expression and inhibition of tumor growth [147,148,149]. On the other hand, pseudogenes can also act as oncogenes in the body. Karreth et al. [150] reported that BRAF P1, a pseudogene of the oncogene BRAF, is mainly expressed in tumor cells. Importantly, BRAF P1, with multiple miRNA binding sites, can regulate the expression of its parental gene by sequestering specific miRNAs through a competitive endogenous RNA mechanism, and this specificity depends on the tumor type. Hao et al. [151] showed that the pseudogene AKR1B10P1 can promote the growth and motility of HCC cells in vitro and in vivo. Further studies found that the pseudogene AKR1B10P1 effectively abolished miR-138-induced SOX4 mRNA degradation and enhanced EMT in HCC by directly sponge miR-138.

To date, although substantial evidence suggests that pseudogenes play an important role in cancer, their functions and underlying mechanisms remain largely undetermined. One of the main functions of pseudogenes has been found to act as miRNA decoys, competing with miRNA that may target parental genes and regulating their activities, thereby playing a role in promoting or suppressing tumors.

## 8. Extracellular Secretion and Endocytosis of miRNA

Notably, extracellular vesicles (EVs) as communication mediators with neighboring or distant cells undoubtedly provide a new pathway for differential expression and functional regulation of miRNA in the tumor and tumor microenvironment. These cells rely on the active transport system of EVs to deliver their own synthesized miRNAs to recipient cells, where they act as endogenous miRNAs to regulate multiple target genes or signaling pathways [152,153]. For example, in a hypoxic environment, lung cancer cells receive EV-derived miR-31-5p to activate MEK/ERK signaling, thereby contributing to their development and metastasis [154]. Besides, tumor cells secrete produced miRNAs to promote their own growth and create a tumor microenvironment that evades surveillance and attack by the immune system [155,156]. In particular, miR-1298-5p secreted by glioma cells not only reduces the level of endogenous miR-1298-5p to promote their proliferative ability but also activates myeloid-derived suppressor cells (MDSCs) through the NF-κB pathway to suppress the immune system [157]. These studies have shown that miRNA expression is highly regulated by the cell’s own secretion, and when the host receives the entry of exogenous miRNA, it can also cause significant changes.

Despite the abundance of miRNA species, some miRNAs involved in cancer proliferation, growth, and metastasis were significantly enriched in extracellular vesicles, suggesting the existence of a mechanism that regulates the loading of specific miRNAs into vesicles for exocrine expression [158,159]. Given the variety of miRNAs in EVs and the specificity of different tumor cell biological behaviors and states, it remains mysterious whether these “miRNA weapons” employed by tumor cells have their specific screening and secretion mechanisms. Fortunately, some recent studies have reported that the molecular mechanism of miRNA screening and loading in tumor cells may be related to different heterogeneous nuclear ribonucleoproteins (hnRNPs). First of all, in lung cancer cells, hnRNPA2B1 binds the tumor suppressor miR-122-5p through the EXO motif to selectively sort and transfer it into secreted EVs [160]. Meanwhile, the delivery of lung cancer EVs-miR-122-5p promoted hepatocyte migration by increasing the expression of N-cadherin and vimentin, which plays an important role in the establishment of the pre-metastatic microenvironment in lung cancer and liver metastasis [160]. Furthermore, in highly metastatic cancers, another heterogeneous nuclear ribonucleoprotein (hnRNPK), driven by non-caveolin-1 (CAV1), assists the EV loading and secretion of miR-148a-3p, which favors migration of prostate and colorectal cancer cells [161].

At present, these studies expand the understanding of the expression regulation of tumor miRNA, including changes in endogenous miRNA caused by active cellular secretion and host acceptance of miRNA. However, whether or not the extracellular secretion of miRNA is highly selective and related underlying mechanism remains to be fully elucidated.

## 9. Conclusions and Perspectives

Up to now, many studies have demonstrated that the abnormal expression of miRNA in tumors is closely related to the occurrence and development of tumors. Therefore, the detailed elucidation of miRNA expression regulation mechanisms is of great significance for the diagnosis, prognosis, and treatment of cancer patients.

Current research on the regulation of miRNA expression mainly focuses on multiple processes ranging from gene transcription to post-transcriptional modification regulation. However, the mechanisms underlying the uncontrolled expression of miRNA in tumors are far from clear. We propose there are three main facets of the regulatory mechanism of miRNAs in tumors that still need to be further elucidated. Firstly, recent studies have shown that the gut microbiota can regulate the expression of host miRNAs to participate in tumorigenesis and development [162,163]. In addition, plant-derived miRNAs in food can be taken up by mammals and participate in the regulation of miRNA composition and function in vivo [164,165]. More importantly, intermittent fasting can also affect the expression of related miRNAs and shows amazing anticancer effects [166,167,168]. These studies suggested that human diet, living habits, and other aspects might can affect the expression of body-related miRNAs and jointly promote or inhibit tumorigenesis. Secondarily, in fact, miRNA, as a kind of powerful, numerous, and abundant small RNA, has complex and diverse regulatory factors involved in its production. Therefore, how to outline the underlying connections among these different factors at distinct levels is still urgent and important for understanding the regulatory mechanisms of miRNA. Thirdly, the expression of miRNA has temporal and spatial specificity in different tissues or cells, and its expression regulation mechanisms are often different. Therefore, in the future, it is necessary to use advanced technologies such as single-cell multi-omics sequencing and high-resolution spatial omics to further explore the specific regulatory mechanisms of miRNA in specific tumors.

In conclusion, it is necessary to comprehensively study the complex network covering the expression of miRNAs and their regulatory mechanisms, so as to open up a new way for human beings to deeply understand the expression and regulation mechanisms of tumors and to apply small molecular means such as miRNA to prevent and treat tumors.

## Figures and Tables

**Figure 1 cells-11-02852-f001:**
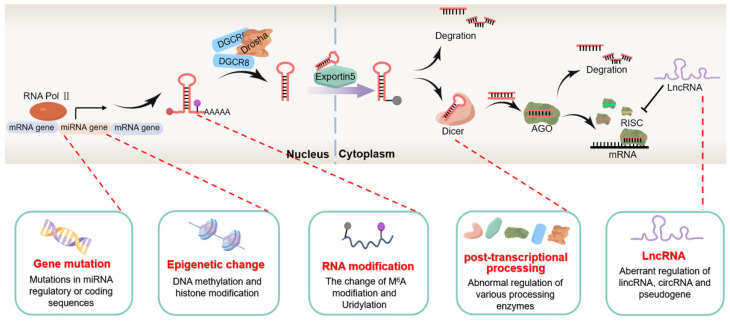
Summary of miRNA expression regulation mechanism in tumors. The expression of miRNAs in tumors is complexly regulated by miRNA genome mutations, epigenetic changes, RNA modifications, abnormal splicing by processing enzymes, and lncRNAs.

**Figure 2 cells-11-02852-f002:**
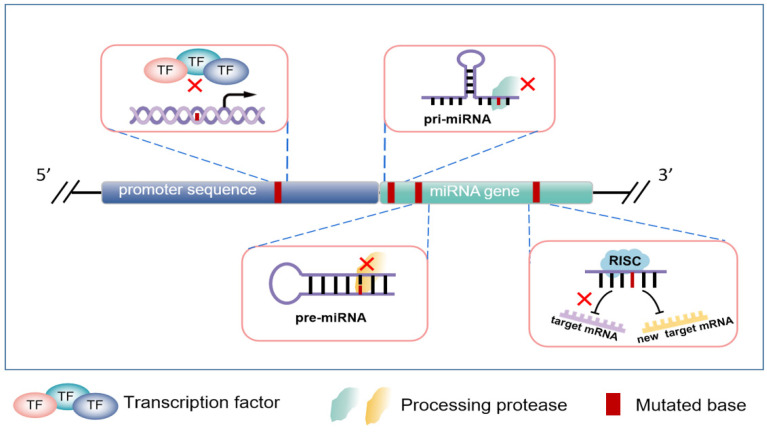
The effect of miRNA genomic mutation in tumors. Base mutation in promoter sequence can affect the binding ability of related transcription factors or change their types. In addition, base mutation in the initial or precursor sequence of miRNA can lead to mislocation and shearing of splicing enzymes. More importantly, base mutation presents in mature miRNA sequence may weaken their inhibitory effect or retarget a new set of mRNAs.

**Figure 3 cells-11-02852-f003:**
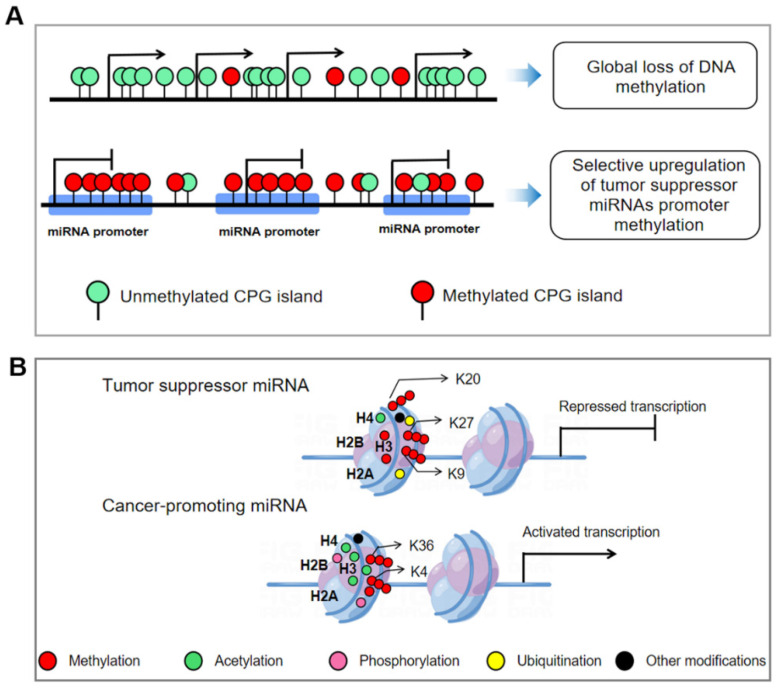
Epigenetic modification of miRNA genomes in tumors. (**A**) Global loss of DNA methylation in tumor cells and selective hypermethylation of CPG islands in tumor suppressor miRNA promoters. (**B**) Histones near the tumor suppressor miRNA in tumor cells show an enrichment of numerous transcriptional silencing-related markers (e.g., H3K9me3, H4K20me3, and H3K27me3) and a significant decrease of transcriptional activation markers (e.g., acetylation of H3 and H4 lysine residues, H3K4me3 and H3K36me3), while histones near cancer-promoting miRNAs showed the opposite labeling state. Moreover, histone modifications involve a large number of underappreciated types that together regulate the tightness of histones.

**Figure 4 cells-11-02852-f004:**
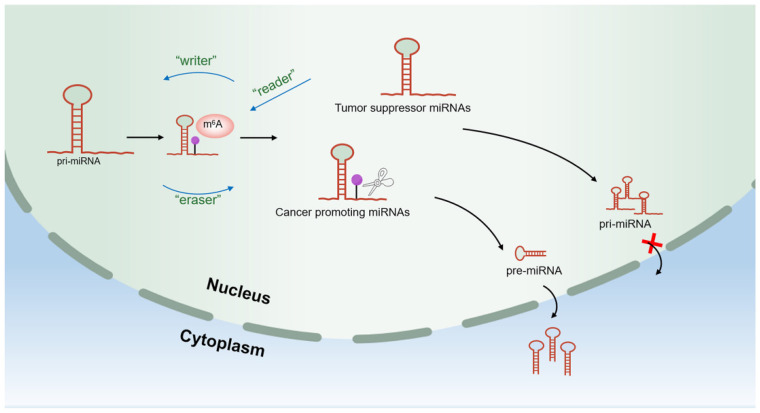
The m6A modification regulation of miRNAs in tumors. The m6A modification process of miRNA is coordinated by m6A writers, readers, and erasers and determines its maturation and expression. Purple circle indicates m6A modification performed by m6A writers at the pri-miRNA motif, which facilitates processing by associated proteases and favors the production of pre-miRNA. However, tumor suppressor miRNA cannot be efficiently labeled by m6A due to lack of m6A writers or readers, or over-processing by demethylases, which results in a large number of pri-miRNAs remaining in the nucleus and unable to undergo normal processing steps. On the contrary, oncogenic miRNA is more easily modified by m6A to promote its expression.

**Figure 5 cells-11-02852-f005:**
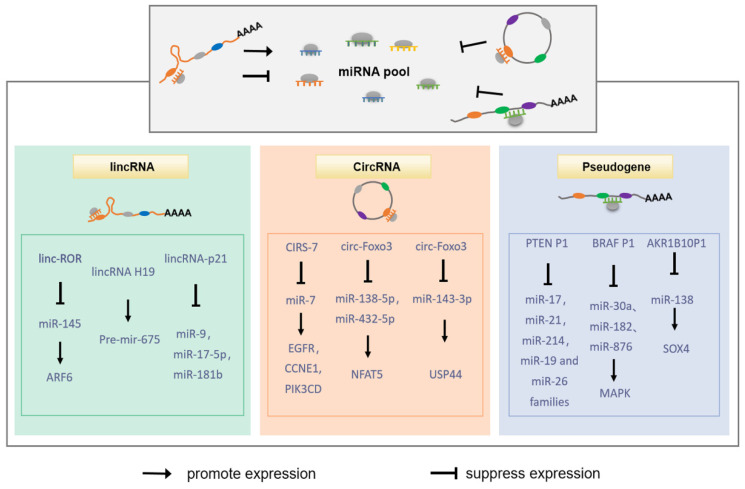
Regulation of miRNA by LncRNAs. LincRNAs, circRNAs, and pseudogenes can act as “sponges” of miRNA through the ceRNA mechanism, thereby participating in the regulation of specific miRNA targeting mRNA in different tumor cells.

**Table 1 cells-11-02852-t001:** The effect of abnormalities in miRNA processing-related proteins.

Changes	Changed Target miRNAs	Cancer Types	Outcome	References
DROSHA				
Upregulation	MiR-31; miR-126, etc.	Squamous cell carcinoma; NSCLC	Increased tumor cell viability and invasion; positive association with poor prognosis	[93,94]
Downregulation	Global miRNA expression	Breast cancer	Positive association with older age at diagnosis, higher histological grade, higher tumor size, and metastasis	[95]
DROSHA RNase domain mutation	Let-7 family; miR-200 family	Wilms tumor	Positive association with a higher rate of relapse and death	[96,97]
XPO5				
Upregulation	miR-21, miR-10b, miR-27a, miR-92a, miR-182, etc.	Colorectal cancer	Positive association with worse clinicopathological features, and poor survival	[98]
Downregulation	MiR-433, miR-22, etc.	Cholangiocarcinoma	Increased cell proliferation and shorter cilia	[99]
Frameshift mutations	MiR-200 family, let-7a, miR-26a, etc.	Endometrial cancer, colorectal cancer, stomach cancer	Increased tumor cell growth and colony-forming capacity	[100]
DICER				
Upregulation	Global miRNA expression	Prostate cancer	Positive association with clinical stage, lymph node status, and Gleason score	[101]
Downregulation	Let-7; miR-1914-5p and miR-541-5p	Lung cancer; cholangiocarcinoma	Positive association with patient survival; increased tumor cell proliferation and invasion	[102,103]
DICER RNase domain mutation	Let-7 family	Wilms tumors; ovarian Sertoli-Leydig cell tumors	Define a distinct subclass of Wilms tumors; increased ovarian oncogenic transformation	[96,104]
Argonaute				
Upregulation	MiR-148a-3p, miR-361-5p, miR-15b-5p, etc.	OC	Positive association with advanced FIGO stage, lymph-node metastasis, poor survival rate	[105]
Downregulation	MiR-185-3p, miR-223, miR-150, etc.	CRC, aggressive breast cancers	Elevated metastatic capacity of CRC and breast cancer	[106,107]
DGCR8				
Upregulation	MiR-27b, miR-32, miR-106b/25 cluster, miR-30c-1, miR-15b, miR-16-2 and miR-153-2	OC, prostate cancer	Increased cell proliferation, migration, invasion, and drug resistance; increased prostate tumor cell proliferation	[108,109]
DGCR8 domain mutation	MiR-29c, miR-30e, miR-100, miR-221, miR-125a, etc.	Schwannoma, follicular thyroid carcinomas	Positive association with poor prognosis	[110,111]

## Data Availability

Not applicable.

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
