# Peer review of "Mechanisms Controlling MicroRNA Expression in Tumor"

_cells, 2022, doi:10.3390/cells11182852_

Round 1

Reviewer 1 Report

The review article „Mechanisms Controlling MicroRNA Expression in Tumor” is focused on the causes and mechanisms of aberrant miRNA expression in human tumors.

There is a lot of review articles summarizing the changes of miRNA expression and its consequence in tumors but a comprehensive reviews on the mechanisms of aberrant miRNA transcription are lacking. Authors focused on important issue.

The scope of the article is relatively large. The article describe role different mechanisms including genetic, transcriptomic and epigenetic changes.

Illustrations are clear and useful.

In my opinion this article is worth publishing.

Reviewer 2 Report

 This is a very comprehensive review of the regulation of miRNA expression in cancer.  The authors describe various regulatory mechanisms and refer to the current literature.  The illustrations are clear and informative. Most of my comments concern editorial issues and I have addressed only the most obvious ones. The authors should therefore carefully revise the text of the manuscript

 A. Comments  for editorial changes

The authors should not only change the mentioned points but also overwork the whole text carefully.

1. The heading 3.1 "MiRNA promoter mutation" should be changed to "Mutations in promoter regions of miRNA encoding genes"

2. The heading "3.1 Variation of miRNA coding sequence" should be renumbered to 3.2 and the heading text should be changed to "Variations of miRNA coding sequences".

 3. The authors should change the title of chapter 4 to ". Regulation of miRNA by epigenetic modifications".

4. The subchapters of chapter 4 need to be renumbered and the subheadings  4.1. etc. should be changed from "Histone modification and miRNA" to "The influence of histone modification on miRNA expression"  and so on.   In addition, the headings should also be changed in chapter 5, accordingly.

5. Acetylation is already a modification, so please change "4.2.2. histone methylation modification" to "histone methylation". Please change "4.2.3. other covalent modification of histone" to "other modifications of histones". 

6. According to the guidelines for Latin sub-terms, "miRNA" should not be written as "MiRNA" at the beginning of a sentence, but as "MicroRNA".

B.  Secretion  of miRNA

The expression of miRNA also depends on the active secretion of miRNA.   The authors should at least  add a short chapter  that endogenous levels of miRNA are highly regulated by secretion processes.

Reviewer 3 Report

I must congratulate the authors for an exhaustive review on miRNAs, specifically the detailed description on mechanisms that regulate miR expressions. 

A table of studies cited with respect to variation in miR levels or regulatory mechanisms impacting its expression and their clinical consequence will add more relevance. At this time it is very descriptive and adding some discussion on impact of risk or on response would make it more appealing for the readers

Round 2

Reviewer 2 Report

The review was overworked vey careful and the authors meet all my concerns.  It  is a very comprehensive  and informative manuscript about the regulation of miRNA .  I am sure that it will find  broad interest  in the audience.